# Bio-Inspired Self-Healing, Shear-Thinning, and Adhesive Gallic Acid-Conjugated Chitosan/Carbon Black Composite Hydrogels as Suture Support Materials

**DOI:** 10.3390/biomimetics8070542

**Published:** 2023-11-12

**Authors:** Hyun Ho Shin, Ji Hyun Ryu

**Affiliations:** 1Department of Chemical Engineering, Wonkwang University, Iksan 54538, Jeonbuk, Republic of Korea; tlsgusgh1231@wku.ac.kr; 2Department of Carbon Convergence Engineering, Wonkwang University, Iksan 54538, Jeonbuk, Republic of Korea; 3Smart Convergence Materials Analysis Center, Wonkwang University, Iksan 54538, Jeonbuk, Republic of Korea

**Keywords:** gallic acid-conjugated chitosan, carbon black, suture support, self-healing, shear-thinning, tissue adhesives

## Abstract

The occurrence of leakage from anastomotic sites is a significant issue given its potential undesirable complications. The management of anastomotic leakage after gastrointestinal surgery is particularly crucial because it is directly associated with mortality and morbidity in patients. If adhesive materials could be used to support suturing in surgical procedures, many complications caused by leakage from the anastomosis sites could be prevented. In this study, we have developed self-healing, shear-thinning, tissue-adhesive, carbon-black-containing, gallic acid-conjugated chitosan (CB/Chi-gallol) hydrogels as sealing materials to be used with suturing. The addition of CB into Chi-gallol solution resulted in the formation of a crosslinked hydrogel with instantaneous solidification. In addition, these CB/Chi-gallol hydrogels showed enhancement of the elastic modulus (G′) values with increased CB concentration. Furthermore, these hydrogels exhibited excellent self-healing, shear-thinning, and tissue-adhesive properties. Notably, the hydrogels successfully sealed the incision site with suturing, resulting in a significant increase in the bursting pressure. The proposed self-healing and adhesive hydrogels are potentially useful in versatile biomedical applications, particularly as suture support materials for surgical procedures.

## 1. Introduction

Suturing is a critical step in many surgical procedures [1,2,3]. Although suturing techniques are well established, tissue injury due to excessive suturing, localized inflammation, bacterial infections, and leakage around the suture have often been reported [1,2,3]. More importantly, leakage from the sutured regions occasionally causes serious complications [4,5,6,7,8,9,10,11]. For instance, anastomotic leakage after colorectal surgery can lead to the serious complications of abdominal and pelvic sepsis, wound infections, fecal incontinence, and tumor recurrence, as well as postoperative death [9,10,11]. The overall incidence of anastomotic leakage after colorectal cancer resection is reported to be 2–19% [12,13,14]. In general, the bursting pressures of the gastrointestinal anastomosis sites are significantly decreased during the first three days, and most leakages after colorectal surgery occur within the first week [15,16]. Fibrin glue, chitosan patches, collagen patches, and polyglycolic acid meshes have been developed for the prevention of anastomotic leakages [17,18,19,20,21,22]. For instance, fibrin-based tissue adhesives show increases in anastomotic bursting pressures with promotion of fibrous healing, resulting in successful prevention of anastomotic leakage in several experimental and animal studies [23,24]. However, it remains a challenging issue owing to variety in surgical settings and patient medical histories. Thus, development of suture support materials to prevent leakage from anastomosis sites in various surgical settings is highly desirable.

Chitosan is a naturally occurring polysaccharide that is one of the promising materials for various biomedical applications, such as tissue adhesives, wound dressing/healing materials, wrinkle fillers, hemostats, drug delivery depots, and tissue engineering scaffolds, due to its biocompatibility, biodegradability, and tissue-adhesive properties [25,26,27,28,29]. The primary amine groups of chitosan exposed via partial deacetylation of chitin allow the adhesion to tissue surfaces through electrostatic interactions and hydrogen bonds [28]. However, tissue-adhesive properties with the bioactivity of chitosan are limited due to its poor solubility in neutral physiological conditions [28]. Chitosan derivatives with various functional groups including carboxylic acid, sulfonic acid, thiol, phenol, catechol, or gallol groups are synthesized to increase its solubility and/or biological activity in aqueous solution [30,31,32,33,34,35,36]. For instance, thiol-containing chitosan shows increased water solubility with improved muco-adhesive properties [34,35,36]. Thus, chemical conjugations of functional groups to chitosan backbones can extend the range of applications in biomedical fields.

Inspired by nature, polymeric tissue adhesives created by conjugating phenolic moieties (i.e., phenol, catechol, and gallol groups) to polymers have been extensively developed for various biomedical applications [28,37,38,39,40]. The conjugation of catechol or gallol moieties to polymeric backbones provides beneficial functionalities along with strong tissue adhesiveness [28,37,38,39,40]. For instance, catechol-conjugated chitosan (Chi-catechol), one of the mussel-inspired polymers, shows enhanced solubility in aqueous solution owing to the solubility of catechol moieties and hydrodynamic diameter changes [40]. Also, Chi-catechol with various physical states of solutions, hydrogels, patches, and films exhibits excellent tissue-adhesive properties [41,42,43,44]. Notably, Chi-catechol patches are effective in preventing anastomotic leakage after colorectal surgery by robustly sealing the anastomosis sites in animal models [18]. In addition, gallol-functionalized polymers inspired by plants also show similarly strong tissue-adhesive properties to catechol-functionalized polymers [45,46,47,48,49,50]. Gallol-containing chitosan (Chi-gallol, 10 wt%) hydrogels exhibit strong adhesiveness to porcine skins in the presence of metal ions (Fe^3+^) or strong oxidants (NaIO_4_) through electrostatic interactions, hydrogen bonds, cation-π interactions, and subsequent oxidations of remnant pyrogallol functional groups [45]. Chi-gallol films also show tissue adhesiveness with self-wrapping properties, resulting in hemostasis in bleeding mouse liver models [49]. In addition, double-layered patches composed of Chi-gallol and hyaluronic acid successfully prevent anastomotic leakage by sealing anastomosis sites via Chi-gallol-mediated adhesions in animal models [50]. Thus, phenolic-compound-conjugated polymers have remarkable potential as suture support materials.

Carbon nanomaterials including graphene, carbon nanotubes, and carbon black (CB) nanoparticles can be used to enhance the mechanical properties of polymeric biomaterials as additives [51,52]. For instance, electrospun nanofibrous scaffolds composed of CB nanoparticles, polyacrylonitrile (PAN), and hydroxyapatites show a significant increase in the tensile strengths and Young’s modulus compared to PAN nanofibers [52]. In addition, CB-containing polycaprolactone (PCL) scaffolds exhibit a Young’s modulus of 101.25 MPa after the addition of CB (7 wt%), which is higher than that of PCL scaffolds (61.80 Mpa) [53]. Also, elastic modulus values of CB-incorporated phenylboronic-acid-conjugated alginate and polyvinyl alcohol composite hydrogels are increased as a function of CB concentration [54]. The toxicity of CB nanoparticles is known to depend on the various factors, such as physical states, chemical compositions, and experimental conditions [55,56,57]. For instance, toxic substances absorbed during CB synthesis significantly increase the toxicity [55]. In addition, the large surface areas of CB nanoparticles can also increase the toxicity [55]. Furthermore, CB nanoparticles can induce cell necrosis and subsequent inflammatory responses [58]. However, several methods to reduce the cytotoxicity of CB through dispersion of CB, surface modification of CB, or formation of CB/polymer complexes have been reported [52,59,60]. Thus, rational design of CB-containing materials can increase the mechanical properties of the polymeric materials with biocompatibility.

We hypothesized that applying suture support materials prepared with strong mechanical and tissue-adhesive properties using gallol-containing polymers could enhance bursting pressures at anastomosis sites. In this study, we developed carbon-black-containing gallic acid-conjugated chitosan (CB/Chi-gallol) hydrogels as suture support materials. The synthesis of Chi-gallol was confirmed through ^1^H NMR and UV-Vis spectroscopic studies. The gelation behaviors and rheological properties of the CB/Chi-gallol hydrogels were monitored. The in vitro cell viability of the extracts of the CB/Chi-gallol hydrogels was evaluated. In addition, the in vitro tissue adhesiveness of CB/Chi-gallol hydrogels was measured using a modified lap shear test. More importantly, in vitro leakage models around sutures with one, two, or three suture stitches of 4-0 vicryl suture were prepared after porcine intestine was cut to 1 cm in a length. The hydrogels were applied to the incision sites with the suturing. The bursting pressures were measured, and a sudden decrease in pressure caused by air leakage was monitored to evaluate the effectiveness of CB/Chi-gallol hydrogels as suture support materials.

## 2. Materials and Methods

### 2.1. Materials

Chitosan (medium molecular weight, 200–800 cp), gallic acid (GA), and sodium chloride (NaCl) were purchased from Sigma-Aldrich (St. Louis, MO, USA). 1-Ethyl-3-(3-dimethylaminopropyl)carbodiimide hydrochloride (EDC) was purchased from TCI-SU (Tokyo, Japan). Carbon black CB (30 nm) was purchased from the Graphene Supermarket (Ronkonkoma, NY, USA). Ethanol (Daejung Chemical, Siheung, Republic of Korea) was used as received without further purification. All the other chemicals were of analytical grade.

### 2.2. Synthesis of Chi-Gallol

Chi-gallol was synthesized by conjugating GA to a chitosan backbone. First, chitosan (1 g) was hydrated with 1N HCl solution (10 mL), and distilled and deionized water (DDW; 89 mL) was added to the chitosan solution. After homogeneous dissolution of chitosan, the pH was adjusted to 5.0. Subsequently, GA (1.1 g) and EDC (1.2 g) in a co-solvent of (25 mL) of ethanol and DDW (1:1 *v*/*v*) were added to the chitosan solution and allowed to react for 12 h. The pH of the reaction solution was maintained between 4.5 and 5.5 during the reactions. The synthesis of Chi-gallol was performed at room temperature. The product was purified via membrane dialysis (MWCO = 3.5 kDa, SpectraPor) against a pH 2.0 NaCl solution (10 mM) for 2 days and DDW for 4 h and was then lyophilized.

### 2.3. ^1^H NMR and UV-Vis Spectroscopic Studies of Chi-Gallol

The synthesis of Chi-gallol was confirmed through ^1^H NMR (Nuclear Magnetic Resonance) and UV-Vis (ultraviolet-visible) spectroscopic studies. Chi-gallol was dissolved in deuterium oxide (D_2_O) at a concentration of 2 mg/mL. The ^1^H NMR spectra of Chi-gallol were obtained using ^1^H NMR spectroscopy (Bruker Avance III, 500 MHz, Billerica, MA, USA). In addition, Chi-gallol was dissolved in DDW at a concentration of 0.025 mg/mL. The UV-Vis spectra of the Chi-gallol were obtained using a (UV/Vis) spectrophotometer (UV-2450, Shimadzu, Kyoto, Japan) at the Core Facility for Supporting Analysis & Imaging of Biomedical Materials of Wonkwang University, supported by the National Research Facilities and Equipment Center. To calculate the gallol substitution rates of Chi-gallol, the GA solutions (0.001, 0.005, 0.01, 0.025, 0.05, and 0.1 mg/mL) were used to prepare the standard curves between the absorbance at 265 nm and GA content. The degree of gallol substitution was calculated by comparing the absorbance of Chi-gallol at 265 nm with standard curves of GA concentrations.

### 2.4. Preparation of CB/Chi-Gallol Hydrogels

CB/Chi-gallol hydrogels were prepared by adding CB into the Chi-gallol solutions. Chi-gallol (1, 2, and 4 wt%, respectively) was dissolved in pH 7.0 PBS solution. After the complete dissolution of the Chi-gallol, CB was added into the solution. The concentration of CB was varied from 2 to 8 wt%. The final concentrations of the CB/Chi-gallol hydrogels were fixed at 4 wt% Chi-gallol and 8 wt% CB. The preparation of CB/Chi-gallol hydrogels was performed at room temperature.

### 2.5. Rheological Analysis of CB/Chi-Gallol Hydrogels

To monitor the viscoelastic behaviors of the CB/Chi-gallol hydrogels, a rotating rheometer (Kinexus Lab+, Netzsch, Germany) equipped with a 20 mm parallel plate was used. For the frequency sweep measurements, the sweep of the frequency was varied from 0.1 to 10 Hz. The elastic modulus (G′) and viscous modulus (G″) values of the CB/Chi-gallol hydrogels with different CB concentrations were monitored at the constant strain (1%) as a function of frequency. In addition, the temperature was fixed at 25 °C during the frequency sweep measurements. To monitor the changes in the G′ values of the CB/Chi-gallol hydrogels, the hydrogels with various concentrations of CB and Chi-gallol were prepared. After homogeneous mixing, the G′ values were measured at a frequency of 1 Hz. To evaluate the recovery of the hydrogels, step–strain measurements were performed. The G′ and G″ values were monitored with continuous strain changes (0.5%, 300%, 0.5%, 300%, and 0.5%), each applied for 120 s. Shear-thinning properties of the hydrogels were measured using a 40 mm cone and plate geometry. The viscosity (Pa.s) changes of the hydrogels were monitored as a function of shear rate (1/s). All measurements were performed in triplicate.

### 2.6. Morphological Analysis of CB/Chi-Gallol Hydrogels

The morphology of the CB/Chi-gallol hydrogels was analyzed using scanning electron microscopy (SEM, S-4800, Hitachi Ltd., Tokyo, Japan) at an acceleration voltage of 5 kV at the Core Facility for Supporting Analysis & Imaging of Biomedical Materials at Wonkwang University, which is supported by the National Research Facilities and Equipment Center. Briefly, CB/Chi-gallol hydrogels were prepared with 8 wt% CB concentration and subsequently lyophilized. After fully drying, cross-sectional SEM images of CB/Chi-gallol sponges were obtained. Chi-gallol was used as a control. The samples were coated with platinum before the SEM images were obtained.

### 2.7. Cell Viability of Extracts of CB/Chi-Gallol Hydrogels

To evaluate the cytotoxicity of CB/Chi-gallol hydrogels, cell viability tests were performed following the guideline ISO 10993-5 using the extracts of the hydrogels. Briefly, the hydrogels (0.2 mL) were pre-cured at predetermined time intervals (0, 3, 6, 12, and 24 h) and media (1 mL) were subsequently added onto the hydrogels. After the hydrogels were incubated at 37 °C in a humidified 5% CO_2_ atmosphere for 72 h, the extracts (1 mL) from the hydrogels were obtained. L929 mouse fibroblasts were used for cell viability testing. The cells were seeded onto 96-well plates (10,000 cells/well) and incubated for 24 h. After removing the media, the extracts (200 μL) of the hydrogels were added to the well plates. After 24 h, MTT solution was added to the well plates, and the plates were incubated for 2 h at 37 °C in a humidified 5% CO_2_ atmosphere. The absorbance of the well plates at 570/650 nm was monitored using a microplate reader (BioRad, Model 550, Hercules, CA, USA) to quantify the cell viability. All measurements were performed in triplicate.

### 2.8. Tissue-Adhesive Properties of CB/Chi-Gallol Hydrogels

The tissue adhesiveness of the CB/Chi-gallol hydrogels was measured using a universal testing machine (UTM, Instron 5583, Instron, Norwood, MA, USA) with a 50 N load cell using a modified lap shear test, as previously reported [54]. Briefly, fresh porcine intestinal tissues (Bucknam butcher shop, Iksan, Republic of Korea) were cut into 1 × 1 cm^2^ squares and attached to the edges of polyethylene terephthalate (PET) films (1 × 5 cm^2^). After preparing the porcine-intestine-attached PET films, two intestine tissues on PET films were overlapped by 1 × 1 cm^2^. The CB/Chi-gallol hydrogels were injected between the intestinal tissues. After 1 min of stabilization, the tensile strengths were measured by pulling the probe with a crosshead speed of 1 mm/min. All measurements were performed in triplicate.

### 2.9. In Vitro Bursting Pressure Measurements

Bursting pressure monitoring devices composed of plastic containers, indicators, pressure transmitters, and recorders were used to measure the bursting pressure changes before and after applying the CB/Chi-gallol hydrogels. The center of the intestine was cut to a length of 1 cm using scissors, and CB/Chi-gallol hydrogels were applied to the incision sites between the tissues. Then, one/two/three stitches of 4-0 vicryl suture with interrupted suturing were applied. The pressure was monitored while air was being blown into the container, and the bursting pressure was obtained when a decrease in pressure owing to air leakage was observed. For the control group, one/two/three stitches of 4-0 vicryl suture without the hydrogel applications were applied to the incision sites. After that, the pressure was monitored using a method similar to that described above. All measurements were performed in triplicate.

### 2.10. Statistical Analysis

The Shapiro–Wilk test (alpha = 0.05, *n* = 3) was performed to test the normality. Statistical significance was analyzed through one-way analysis of variance (ANOVA) and Tukey’s test for multiple comparison in Prism 8.1.0 software (GraphPad Software Inc., La Jolla, CA, USA). Significance levels were assigned as follows: * *p* < 0.05, ** *p* < 0.01, and *** *p* < 0.001.

## 3. Results and Discussion

### 3.1. Synthesis and Characterizations of CB/Chi-Gallol Hydrogels

To prepare the CB/Chi-gallol hydrogels, Chi-gallol was first synthesized using standard EDC coupling agents by forming amide bonds between the amine groups in the chitosan backbone and the carboxylic acid group of GA (Figure 1a). As shown in Figure 1b,c, the ^1^H NMR and UV–Vis spectra of Chi-gallol were obtained to confirm the conjugation of GA to the chitosan backbones. The gallol protons of Chi-gallol were found at 6.9 ppm in the ^1^H NMR spectra (Figure 1b). As previously reported, the peak at 6.92 ppm in the ^1^H NMR spectra is attributable to the protons of aromatic rings in the GA-conjugated chitosan [61,62]. In addition, an absorbance peak at 265 nm, caused by GA conjugation, was observed in the UV–Vis spectra of Chi-gallol (Figure 1c). The degree of gallol substitution (DOS) of Chi-gallol was calculated to be approximately 13.9% by measuring gallol content in the chitosan backbone using its absorbance at 265 nm against standard curves of GA concentrations.

For the fabrication of CB/Chi-gallol hydrogels, Chi-gallol (4 wt%) was dissolved in pH 7.0 PBS solution, and CB was added to the Chi-gallol solution in varying CB concentrations (Figure 2a). Chi-gallol solution was in a viscous state but became a hydrogel with solidification upon the addition of CB. In addition, the gelation of CB/Chi-gallol hydrogels occurred within 1 min after the addition of CB. As expected, no flow was observed in the CB/Chi-gallol hydrogels (Figure 2b, right), whereas Chi-gallol alone flowed downward (Figure 2b, left).

An increase in CB concentrations in the CB/Chi-gallol hydrogel networks significantly affects the elastic modulus (G′) values of hydrogels. The incorporation of CB into various hydrogels enhanced their mechanical properties, as previously reported [54]. As aforementioned, Chi-gallol alone was a viscous solution. The G′ values of Chi-gallol alone were lower than the G′′ values at all considered frequencies (1–10 Hz) (Figure 3a). However, the G′ values were slightly increased and were higher than those of G″ values in the frequency ranges of 1 to 10 Hz when CB (2 wt%) was added to the Chi-gallol solution (Figure 3b). In addition, further increases in the G′ values of CB/Chi-gallol hydrogels were observed accompanying increased CB concentrations of 4 wt% (Figure 3c), 6 wt% (Figure 3d), and 8 wt% (Figure 3e). The average G′ value of Chi-gallol alone was 66.4 ± 22.2 Pa. After the addition of CB, the average G′ values of CB/Chi-gallol hydrogels increased to 1.0 ± 0.2 kPa for 2 wt% CB, 4.0 ± 0.6 kPa for 4 wt% CB, 5.6 ± 0.6 kPa for 6 wt% CB, and 9.4 ± 1.0 kPa for 8 wt% CB (Figure 3f). Therefore, the final concentrations of Chi-gallol and CB to prepare the CB/Chi-gallol hydrogels as suture support materials were fixed at 4 and 8 wt%, respectively. The rheological analysis of CB/Chi-gallol hydrogels clearly showed that the simple mixing of CB and Chi-gallol solution enhanced the elastic moduli with gelation behaviors.

The morphological analysis of CB/Chi-gallol hydrogels was performed to monitor the changes before and after addition of CB into the Chi-gallol networks. As shown in Figure 4a,b, the morphologies of Chi-gallol and CB/Chi-gallol were relatively similar in SEM images with 100x magnification. Figure 4c shows a magnified view (10 kx) of Figure 4a. The Chi-gallol alone showed smooth surface morphology. However, spread CB particles were found in the CB/Chi-gallol hydrogels at 10 kx magnification (Figure 4d).

In this hydrogel formulation, the concerns about the cytotoxicity due to the presence of CB could be raised. The cytotoxicity of CB was first evaluated as a function of concentration. As shown in Figure 5a, CB alone showed above 80% cell viability of L929 mouse fibroblasts with the concentration ranges from 0.001 to 0.1 mg/mL. However, approximately 30% of cell viability was found at a concentration of 1 mg/mL. In addition, the cell viability of extracts of CB/Chi-gallol hydrogels was measured as a function of curing time (Figure 5b). The extracts from the hydrogels without the curing process showed 67.5 ± 4.8% cell viability. After curing for 3 h, the cell viability was increased to 73.6 ± 1.7%. The extracts from the hydrogels after curing for 6 h showed no significant cytotoxicity, considering that 80–100% cell viability is regarded as non-toxic. This might be due to stably immobilized CB in the Chi-gallol networks during the hydrogelation.

### 3.2. Self-Healing, Injectable, and Adhesive Properties of CB/Chi-Gallol Hydrogels

To verify the self-healing properties of the CB/Chi-gallol hydrogels, air leakage was generated in the hydrogels. As shown in Figure 6a, bubbles (red arrows) in the hydrogels were observed immediately after air leakage occurred. The transparent bubbles disappeared spontaneously over time (Figure 6a). Finally, no bubbles were observed within a few seconds, indicating that the CB/Chi-gallol hydrogels were fully recovered (Figure 6a, last photograph). Step–strain measurements were performed using a rotational rheometer to confirm the self-recovery behavior of the CB/Chi-gallol hydrogels. When a 0.5% strain was applied to the CB/Chi-gallol hydrogels, the G′ values of the hydrogels were 4.2 kPa (Figure 6b). However, the G′ values were significantly reduced to 0.1 kPa immediately after applying a 300% strain. Then, these values were recovered to 4.0 kPa, which was ~94% of these original G′ values. In addition, 83% (3.5 kPa) recovery of G′ values was found in another step, indicating that the hydrogels were reversibly self-healed. Next, the shear-thinning properties of the CB/Chi-gallol hydrogels were monitored to determine their injectability. With an increase in shear rate (0.1 to 100 s^−1^), the viscosity was decreased (Figure 6c). The CB/Chi-gallol hydrogels were injectable without clogging by using 18-gauge needles with a hydrogel injection volume of at least 5 mL (Figure 6d). It was indicated that the hydrogels were injectable using syringe needles and the elastic modulus was recovered after injections.

The tissue-adhesive properties of the CB/Chi-gallol hydrogels were further evaluated using a modified lap shear test for potential applications in suture support. As aforementioned, the porcine intestines were attached to PET films and the tissues were overlapped by 1 × 1 cm^2^. The hydrogels were subsequently applied between two porcine intestines (Figure 7a), and the tensile strength was monitored by pulling the UTM probe. As shown in Figure 7b, the detachment stress of the CB/Chi-gallol hydrogels was 22.3 ± 2.3 kPa, which was considerably higher than that of the control group (0.4 ± 0.2 kPa). In addition, CB (0.1 ± 0.1 kPa), and Chi-gallol (0.4 ± 0.1 kPa) showed no tissue-adhesive properties between the porcine intestines. Thus, these results indicate that CB/Chi-gallol hydrogels can be potentially applied as suture support materials owing to their self-healing and tissue-adhesive properties.

### 3.3. Assessments of CB/Chi-Gallol Hydrogels as Suture Support Materials

The CB/Chi-gallol hydrogels were further used to prevent leakage around the suture. To test the suture support effects of CB/Chi-gallol hydrogels, the center of the porcine intestine was cut to a length of 1 cm by using scissors (Figure 8a, left). Subsequently, one, two, and three stitches of 4-0 vicryl suture with interrupted suturing were applied to the incision sites (Figure 8a, top). For the hydrogel application, the hydrogels were applied to the incision sites, and suturing was performed as described above (Figure 8a, bottom). The bursting pressure of the incision site without the suturing was 2.5 ± 0.9 mmHg (Figure 8b). After suturing with two stitches, the bursting pressure increased to 34.2 ± 1.6 mmHg. When the CB/Chi-gallol hydrogels were applied to incision sites with suturing, the bursting pressure was enhanced to 150.2 ± 13.5 mmHg, which was remarkably higher than that of suturing with Chi-gallol solution application (78.0 ± 12.4 mmHg). Moreover, the bursting pressures were 18.2 ± 3.2 mmHg for one stitch and 82.5 ± 3.2 mmHg for three stitches of suture (Figure 8c). When the hydrogels were applied to incision sites with suturing, the bursting pressures were remarkably enhanced to 79.2 ± 8.4 mmHg for one stitch and 170.2 ± 12.3 mmHg for three stitches, respectively (Figure 8c). These findings suggest that seal-healing and adhesive CB/Chi-gallol hydrogels, as suture support materials, significantly contribute to leakage prevention.

Our study was limited by experimental conditions. Although the leakage mainly occurred at the suture line, the reasons for the leakage are multi-factorial, such as suture techniques, ischemia, excessive tension, and many others [63,64,65,66,67]. In addition, we only focused on the increase in elastic modulus values, tissue-adhesive properties, and bursting pressures of CB/Chi-gallol hydrogels. Therefore, further studies are required to investigate the effectiveness of CB/Chi-gallol hydrogels as suture support materials. Additionally, studies assessing CB/Chi-gallol hydrogels in animal models with anastomosis or preclinical trials should be performed.

## 4. Conclusions

This study demonstrates carbon-black-containing gallic acid-conjugated chitosan hydrogels as suture support materials. Chi-gallol alone was a viscous solution state but became a hydrogel state after the addition of CB. The CB/Chi-gallol hydrogels exhibited increasingly enhanced mechanical properties with increasing CB concentration. In addition, the hydrogels exhibited excellent self-healing, injectability, and tissue-adhesive properties. It was noted that the hydrogels were injectable using 18-gauge needles, adhered to tissue surfaces, and self-recovered after applying external forces. Furthermore, the cytotoxicity evaluation of extracts of the hydrogels after curing for 6 h showed >80% cell viability. Notably, the hydrogels with suturing could seal the incision sites of porcine intestines to reduce air leakage. Thus, CB/Chi-gallol hydrogels can be exploited as suture support adhesive biomaterials for diverse biomedical applications, particularly for preventing leakage at anastomosis sites.

Although the CB/Chi-gallol hydrogels have potential to be explored as tissue adhesives and sealing materials, several challenges still remain for clinical uses. These hydrogels can prevent leakage around sutures by forming physical barriers. The advanced design of suture support materials considering multiple factors associated with the leakage is required. In addition, further examination of the materials is required for clinical applications. For instance, the assessment of the long-term safety and effectiveness of hydrogels is necessary. Also, sterilization methods of the hydrogels can significantly affect their physical and chemical properties. Studies of changes in the physicochemical and biological characteristics according to the various sterilization methods (i.e., gamma irradiation, ethylene oxide sterilizations, and autoclave) are required for clinical applications. Moreover, an approach of adhesive materials as suture support materials is similar to the conventional methods using adhesive materials (i.e., fibrin glue, cyanoacrylate derivatives, chitosan patches, collagen patches, and polyglycolic acid meshes). Thus, novel approaches to prevent leakage around sutures are required.

## Figures and Tables

**Figure 1 biomimetics-08-00542-f001:**
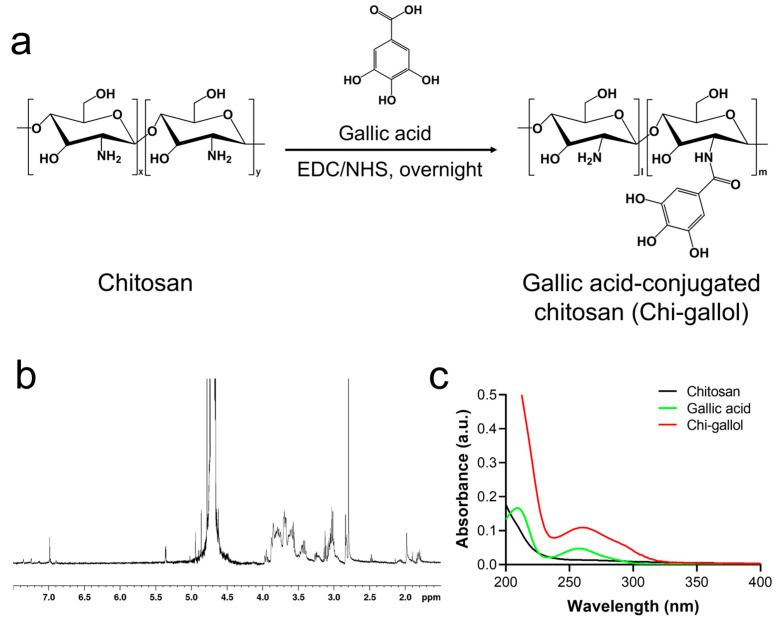
(**a**) Synthesis and chemical structures of gallic acid-conjugated chitosan (Chi-gallol). (**b**) ^1^H NMR and (**c**) UV–Vis spectra of Chi-gallol.

**Figure 2 biomimetics-08-00542-f002:**
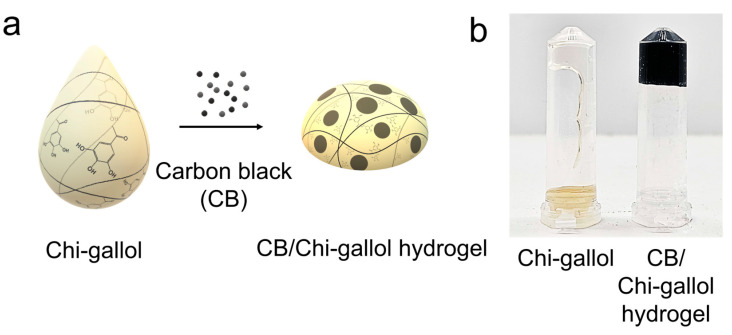
(**a**) An illustration of preparation of carbon-black-containing Chi-gallol (CB/Chi-gallol) hydrogels. (**b**) Photographic images of Chi-gallol solution (**left**) and CB/Chi-gallol hydrogels (**right**).

**Figure 3 biomimetics-08-00542-f003:**
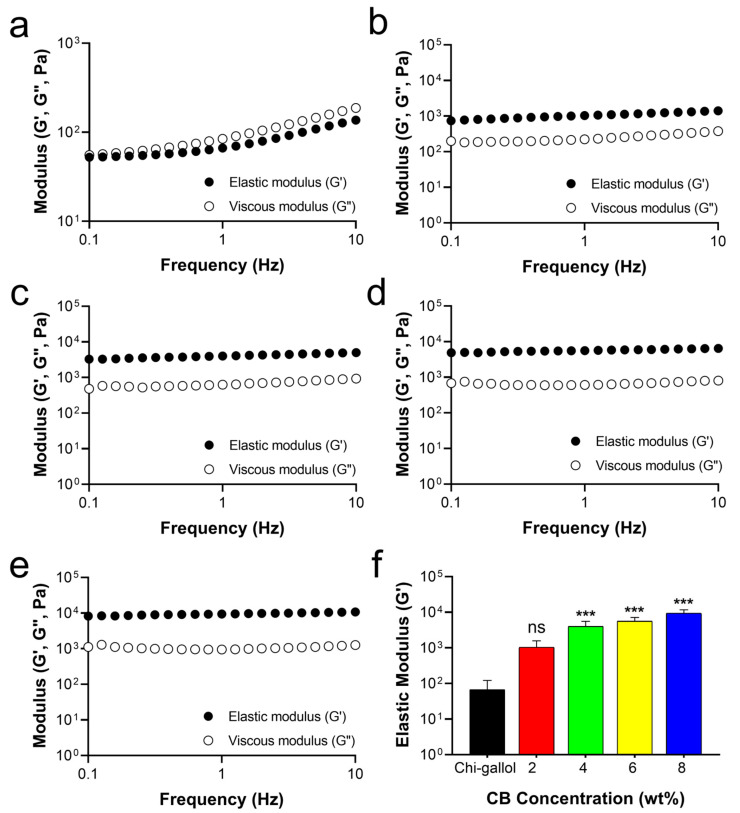
Rheological analysis of Chi-gallol solution and CB/Chi-gallol hydrogels. (**a**–**e**) Frequency sweep measurements of (**a**) Chi-gallol solution, (**b**–**e**) CB/Chi-gallol hydrogels containing various concentrations of CB (2 wt% CB for (**b**), 4 wt% CB for (**c**), 6 wt CB% for (**d**), and 8 wt% for (**e**), respectively). (**f**) Average elastic modulus (G′) values of CB/Chi-gallol hydrogels at 1 Hz frequency as a function of CB concentration. All the rheological measurements were performed at 25 °C. *** indicates a significant difference between changes in elastic modulus (G′) values compared to Chi-gallol solution (*** *p* < 0.001), whereas “ns” indicates not significant (*p* > 0.05).

**Figure 4 biomimetics-08-00542-f004:**
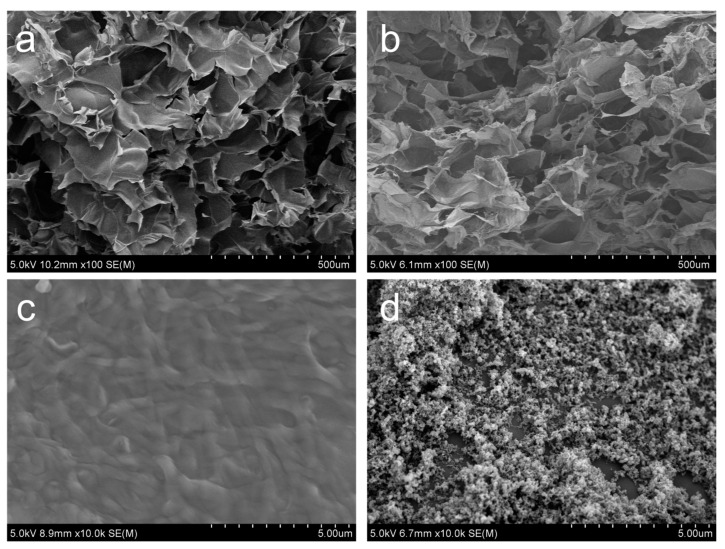
SEM images of (**a**,**c**) Chi-gallol and (**b**,**d**) CB/Chi-gallol hydrogels after lyophilization with 100x and 10kx magnifications.

**Figure 5 biomimetics-08-00542-f005:**
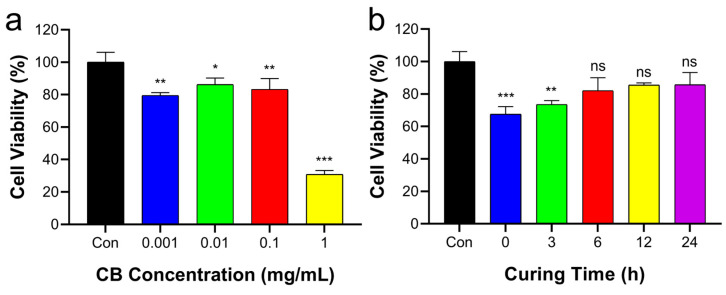
(**a**) Cell viability of CB as a function of concentration (0, 0.001, 0.01, 0.1, and 1 mg/mL). (**b**) Cell viability of the extract of CB/Chi-gallol hydrogels as a function of curing time (0, 3, 6, 12, and 24 h). *, **, and *** indicate statistical significance with respect to the Con (* *p* < 0.05, ** *p* < 0.01, and *** *p* < 0.001, whereas “ns” indicates not significant (*p* > 0.05).

**Figure 6 biomimetics-08-00542-f006:**
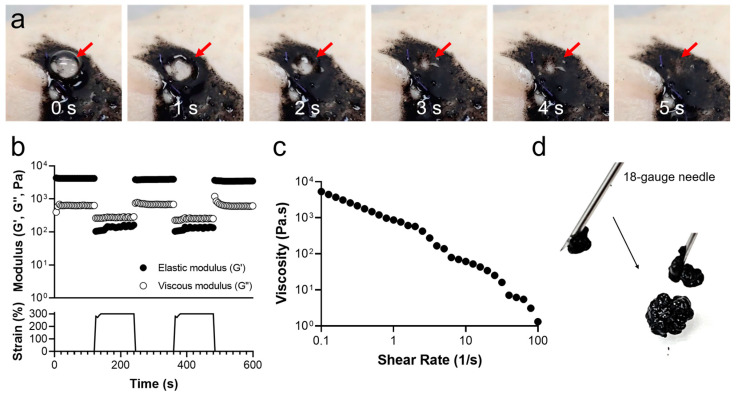
(**a**) Photographs of CB/Chi-gallol hydrogels as a function of time after preparations of air-leakages. (**b**) Step–strain measurements of CB/Chi-gallol hydrogels. (**c**) Viscosity changes of CB/Chi-gallol hydrogels as a function of shear rate. (**d**) Photographic images of CB/Chi-gallol hydrogel injection using 18-gauge needles.

**Figure 7 biomimetics-08-00542-f007:**
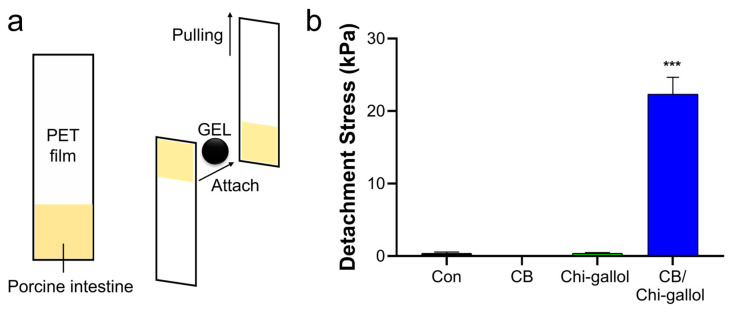
In vitro tissue adhesion studies of the CB/Chi-gallol hydrogels. (**a**) Illustration of tissue adhesion force measurements using UTM. The porcine intestine tissues (1 × 1 cm^2^) were attached to the edge of PET films (1 × 5 cm^2^, first). Two intestine tissues on the PET films were overlapped, and CB/Chi-gallol hydrogels were subsequently applied between the porcine intestines. The tensile strengths were measured by pulling the PET film (second). (**b**) Detachment stresses of control, CB, Chi-gallol solution, and CB/Chi-gallol hydrogels. *** indicates a significant difference between changes in detachment stresses compared to Con (*** *p* < 0.001).

**Figure 8 biomimetics-08-00542-f008:**
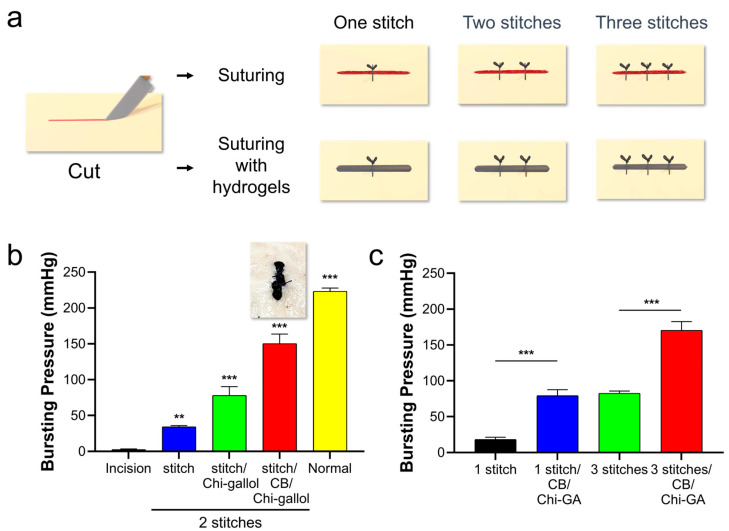
(**a**) Schematic illustrations of bursting pressure measurements. After preparations of 1 cm long incision (first), one, two, and three stitches of 4-0 vicryl suture were applied (top). For the CB/Chi-gallol hydrogel groups, the hydrogels were applied to the incision sites, and suturing was subsequently performed (bottom). (**b**) Measured bursting pressures of intestine with incision, after suturing, after suturing with Chi-gallol solution, after suturing with CB/Chi-gallol hydrogels, and normal tissues. All suturing was performed with two stitches. ** and *** indicate significant differences between changes in bursting pressures compared to Con (** *p* < 0.01 and *** *p* < 0.001). (**c**) Bursting pressure measurements of intestines with incision after one/three stitches of suture without/with the application of the CB/Chi-gallol hydrogels (*** *p* < 0.001).

## Data Availability

The data presented in this study are available on request from the corresponding author.

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
