# Peer review of "Bio-Inspired Self-Healing, Shear-Thinning, and Adhesive Gallic Acid-Conjugated Chitosan/Carbon Black Composite Hydrogels as Suture Support Materials"

_biomimetics, 2023, doi:10.3390/biomimetics8070542_

Round 1

Reviewer 1 Report

Comments and Suggestions for Authors

This article describes the development of self-healing, shear-thinning, tissue adhesive carbon-black-containing gallic acid-conjugated chitosan hydrogels. These systems have been proposed by authors as sealing materials to be used with suturing. According to the findings of this study, the proposed self-healing and adhesive hydrogels are potentially useful in versatile biomedical applications, particularly as suture support materials for surgical procedures. The scope and ideas are very interesting. However, the manuscript contains some limitations that should be addressed.

Comments are outlined below:

1.       Introduction: A literature review on carbon black according to cytotoxicity and use in biomedical applications is mandatory.

2.      Lines 85–96: The results should be removed. At the end of the introduction, the authors should present a brief summary of the methodology followed in this paper.

3.      Sub-section 2.2: What was the synthesis temperature?

4.      Lines 115–121: A new sub-section dedicated to 1H NMR spectroscopy should be added. In addition, add the equation for the calculation of the degree of gallol substitution.

5.      Sub-section 2.3: What was the preparation temperature?

6.      Sub-section 2.4: The authors should confirm that the measurement has been carried out in the linear viscoelastic region (LVR). In addition, the authors should precisely determine the value of either the fixed strain (%) or the fixed stress (Pa) used for the frequency sweep measurements. What was the measurement temperature?

7.      How do the authors sterilize the samples before cell studies? This point is very important and must be highlighted.

8.      Were there any statistical calculations performed on the experiments? For example, ANOVA. If so, they should be discussed. If not, why not?

9.      Figure 3 ABCDEF: The figure legend is unclear. In addition, the measurement temperature should be specified.

10.   Graphs in Figures 3F, 7B, and 7C must be compared to controls according to p levels. What is the significance of the difference between the results according to the p level? Use an ANOVA test, etc.

11.    Figure 5C: There is a possibility to modelize the flow curve of the CB/Chi-gallol hydrogel. By using the Cross or Carreau model etc. This will be very helpful for readers. Thanks to the chosen model, a value of Newtonian viscosity could be easily calculated.

12.    The conclusions section needs improvement, as do the limitations and perspective of this study, which should be clarified and discussed.

Reviewer 2 Report

Comments and Suggestions for Authors

The authors present a highly chemically-based study on the development of novel self-healing hydrogels for use as support to conventional sutures. The methodology is sound as well as the presentation of the physical and chemical data. The novelty of shearthinning gallol-modified hydrogels is not high but the clinical application is relevant. Conceptually, it is relevant to add data pertaining to the biocompatibility of the newly developed material: EDC crosslinking leaves potentially cytotoxic remnants, also gallol is potentially cytotoxic. I would like to see an in vitro cytotox assay showing absence of cytotoxicity.

One thing that is somewhat confusing, is that it would seem that the current hydrogels are an additive to 'normal' sutures: how relevant is that really? How much leakage do the well-skilled surgeons actually leave after stitching wounds?

In conclusion: chemico-physically a sound study with worries about the clinical application.

Round 2

Reviewer 1 Report

Comments and Suggestions for Authors

Geart job. The authors addressed all the concerns. However, this minor should be considered.

-        The graphs in Figure 5 (a and b) must be compared to controls according to p levels. What is the significance of the difference?

Reviewer 2 Report

Comments and Suggestions for Authors

The authors improved the contents of the manuscripts in a satisfactory fashion. The quality warrants publication in Biomimetics

Author Response

We thank the reviewer for the valuable contributions.